# I Think I Should Get Vaccinated, I Feel I Should Not. Individual Differences in Information Processing and Vaccination Behavior (COVID-19)

**DOI:** 10.3390/healthcare10071302

**Published:** 2022-07-14

**Authors:** Cristina Maroiu, Andrei Rusu, Zselyke Pap

**Affiliations:** Department of Psychology, West University of Timisoara, 4 Vasile Pârvan Blvd., 300223 Timisoara, Romania; cristina.maroiu@e-uvt.ro (C.M.); zselyke.pap@e-uvt.ro (Z.P.)

**Keywords:** COVID-19 vaccination, information processing, cognitive reflection ability, thinking style, rationality, experientiality, individual differences

## Abstract

Following the outbreak of the COVID-19 pandemic, the scientific community responded promptly by developing effective vaccines. Still, even though effective vaccines against COVID-19 became available, many people did not seem to be in a rush to become immunized. Community protection can be enhanced if more people decide to vaccinate, and thus it is necessary to identify relevant factors involved in vaccination behavior to find better ways of encouraging it. Vaccination behavior is the result of a decision process that might vary according to individual differences in information processing. We investigated the role of cognitive reflection ability and thinking styles in predicting self-reported vaccination behavior against COVID-19. A sample of 274 Romanian participants was surveyed for the present study, out of which 217 (*M_ag_*_e_ = 24.58, *SD* = 8.31; 53% female) declared they had the possibility to become vaccinated. Results showed that a higher level of cognitive reflection ability significantly increased the odds of becoming vaccinated. A rational thinking style was not linked to vaccination behavior. However, an experiential thinking style indirectly predicted vaccination behavior by means of attitudes towards vaccination. Since individual differences in information processing are, to a certain extent, linked to vaccination behavior, the design of vaccination campaigns could consider that people have specific information needs and address them as such.

## 1. Introduction

While the development of safe and efficacious COVID-19 vaccines provided a way to control the pandemic, vaccine hesitancy slowed the global efforts toward overcoming it. In the era of post-truth, people’s behaviors seem to be shaped more by emotion and personal belief rather than objective facts [1]. Vaccination behavior is preceded by a decision-making process in which people might employ a more or less elaborative processing of information. While the type of processing employed (either less or more elaborative and effortful) may depend on various situational factors (for instance, the amount of time available, the type of information presented etc.), and individual differences may also play an important part. People differ in their ability and motivation to engage in explicit processing and hypothetical thinking. According to ref. [2], there are two potential sources of individual differences in information processing: cognitive reflection ability and thinking style. While cognitive reflection ability is the capacity to sustain decoupled representations for the purpose of inhibition or simulation [3], thinking styles are higher-level regulatory states, such as the tendency to explicitly weigh pluses and minuses before making a decision. The present study focused on individual differences in information processing as potential key pieces for further understanding the complex puzzle of human behavior regarding COVID-19 vaccination. 

In the past decades, research on information processing has been guided by a class of theories, generically called dual process theories. There are several dual-process theories, each having a specific focus [4]. According to ref. [2], all dual process theories share a common ground—they describe two different types of information processing: one that is fast, effortless, and intuitive (system 1), and one that is slow and deliberative (system 2); both types of information processing have advantages and disadvantages. Intuitive processes are helpful, evolutionary tools that provide fast and effortless answers whenever confronted with novel problems. However, when lacking relevant experience, these answers may be inappropriate, and people need to engage in further reflection towards a better-suited response. Studies show that there are individual differences regarding the ability to override the default intuitive response [5]. For instance, in several heuristics and biases experiments, people’s answers vary [2]. When deciding whether to receive the vaccine or not, a deliberative process would involve weighing up the probability of outcomes related to disease and vaccine according to the best available evidence. However, as pointed out by ref. [6], an intuitive perception of risk (system 1) likely plays a major role. This means that the ability to engage in the deliberative processing, necessary to override the default intuitive response, should correlate with the decision to vaccinate. 

Recent findings show that the ability to engage in effortful, deliberative, and reflective processing is negatively associated with the belief that the pandemic is a hoax and positively associated with social distancing and handwashing [7]. In a similar vein, ref. [8] found that people with higher levels of cognitive reflection ability are less likely to rate COVID-19 misinformation as accurate. Additionally, ref. [9] found that working memory capacity, a concept closely related to the cognitive reflection ability [10], predicts social-distancing compliance during the COVID-19. Furthermore, ref. [11] found that cognitive reflection ability is associated with the willingness to be vaccinated against COVID-19, and ref. [12] found that people accepting of a COVID-19 vaccine scored higher on a measure of cognitive reflection ability as compared with people that were hesitant or resistant. Thus, our first expectation is that:

**H1.** 
*Cognitive reflection ability will predict self-reported vaccination behavior (against COVID-19).*


Aside from the ability to override a potentially inadequate intuitive response, an adaptive behavior may also require a disposition to approach problems with a specific thinking style. Thinking styles are defined as trait-like tendencies (i.e., stable across time and context) to use experiential and rational thinking [13]. Thinking style measures tap into one’s goals and epistemic values [2]. In theory, when confronted with a task, a person that has strong tendencies for rational thinking will likely apply cognitive effort and employ an analytic approach, while a person that has strong tendencies for experiential thinking will likely apply a holistic perspective and employ an affective orientation (what feels good) [14,15].

A consistent result confirmed by a meta-analytical investigation [13] showed that the tendency to engage in rational thinking is positively associated with normative decision performance. Although some studies found no link between rational thinking and vaccine uptake [16], others found that a low rational thinking style predicts an engagement in pseudoscientific behaviors meant to protect from COVID-19, like drinking alcohol, and consuming garlic, colloidal silver, or antiviral essential oils [17]. Additionally, a higher rational thinking style was associated with greater compliance with mandated requirements for social distancing at the onset of the COVID-19 pandemic [18]. Thus, we also expect that:

**H2.** 
*A higher level of rational thinking style will predict self-reported vaccination behavior (against COVID-19).*


Experiential thinking implies that people rely on their “gut” feeling [14], which means that they can be seized by their negative emotions [16]. Vaccination has long been an emotionally charged issue, even before the COVID-19 pandemic [19]. When relying on emotions, people appeal to similar prior experiences [13], such as vaccination for common vaccine-preventable diseases. Because vaccination coverage for common vaccine-preventable diseases is high in many countries, people lack the negative experience of the disease. However, experiences with vaccination-side effects are more prevalent; this is likely to shape a negative “gut” feeling [20]. As the authors of ref. [14] explain, experiencing something equals believing it to be true. Since attitudes are overall evaluations of objects, people, behaviors, or policies [21], it is possible that an experiential thinking style might be associated with attitudes toward vaccination. Previous research found that, while forming attitudes, people that show a preference for experiential thinking seek out affective information and are more strongly persuaded by affective messages [22]. Additionally, ref. [23] reported that people with a higher level of experiential thinking rely on affect both in the case of unfamiliar attitude objects and when it comes to familiar-attitude objects. Moreover, the tendency to engage in experiential thinking was found to be associated with vaccine conspiracy beliefs [16] and negatively associated with vaccination attitudes [20]. In a similar vein, ref. [24] showed that a tendency to engage in heuristic processing of information—a concept closely related to experiential thinking style—leads to a higher acceptance of misinformation regarding COVID-19 prevention and treatment. Thus, we expect that:

**H3a.** 
*A higher level of experiential thinking style will predict more negative vaccination attitudes.*


People’s attitudes toward vaccines play an important role in the decision to vaccinate, as single behaviors are tightly related to the compatible measures of attitude [25]. In a study that investigated parents’ decision to vaccinate their children, ref. [26] found that attitude toward vaccines (i.e., confidence in the value of vaccines) is a helpful predictor of parent-reported vaccination behavior. Additionally, ref. [27] found that vaccine attitudes are significantly related to prior vaccination behavior and future intentions to obtain recommended vaccinations. Regarding the vaccination against COVID-19, ref. [28] found that both mistrust regarding the benefits of the vaccine and concerns about unforeseen side effects are linked to the unwillingness to become vaccinated. Taken together, these findings point to the following hypothesis:

**H3b.** 
*Vaccination attitudes predict people’s self-reported vaccination behavior (against COVID-19).*


Previous research indicates that an experiential thinking style should predict vaccine attitudes, and we expect that vaccine attitudes will predict people’s self-reported vaccination behavior (against COVID-19). This is because an experiential thinking style means that people rely on their feelings when forming attitudes. People might form negative attitudes towards vaccines if they mainly rely on the worry or fear evoked by their perceived side effects. In turn, negative attitudes toward vaccines mean that people will be less willing to vaccinate. 

It is less likely that experiential thinking would lead to positive attitudes towards vaccination and, thus, to higher odds of vaccination because vaccination is a counter-intuitive behavior [29]. First, the omission bias shows that people consider it is morally worse to harm someone by doing something than by not doing something [30]. Applied to vaccination, this means that people would anticipate more regret if negative (side) effects happen because of vaccination, as compared with negative effects happening from doing nothing (non-vaccination). Second, the feeling of disgust is an intuitive mechanism that helps us avoid dangerous agents like rotten food or other harmful substances [31]. However, disgust is, to a large extent, dose-insensitive; this means that people can be reluctant to take vaccines although they only contain minute amounts of contaminants [29].

Hence, we expect to find an indirect effect of experiential thinking style on vaccination behavior via its impact on attitudes:

**H3c.** 
*Experiential thinking style indirectly predicts lower odds for vaccination behavior by means of negative attitudes towards vaccination.*


To summarize, the aim of the present study was to investigate a potential link between individual differences in information processing and vaccination behavior. There are two sources of individual differences in information processing: cognitive reflection ability and thinking styles. We expect that both cognitive reflection ability and rational thinking style will directly predict vaccination behavior. Furthermore, we expect that experiential thinking style will have an indirect effect on vaccination behavior through its impact on vaccination attitudes (see Figure 1).

## 2. Materials and Methods

### 2.1. Participants

Students enrolled at the Faculty of Psychology within the West University of Timișoara, Romania, received via e-mail an invitation to participate in this research. They were informed that the research would last approximately 20 min, and they would have to fill in a series of questionnaires. The students were rewarded with course credit for their participation, and if they brought a male guest (also a student/young adult) that would participate in the research, they would receive extra credit (this strategy was used because the students at the Faculty of Psychology are predominantly female, and we aimed to investigate our assumptions on a sex-balanced sample). The final sample was composed of 274 participants (53.6% females; median age = 24.4, SD = 8.2).

### 2.2. Procedure

After signing up for the research, the participants received a Google forms link. The first page of the form presented information about the research, confidentiality of the data, and the participants’ right to withdraw from the research at any point. After offering informed consent, the participants had to offer demographic information, complete measures of vaccine attitudes, answer whether they had access to the vaccine against COVID-19 and, if so, whether they got vaccinated. Then, the participants completed measures for thinking style and cognitive reflection ability.

### 2.3. Measures

Cognitive reflection ability was measured with the Cognitive Reflection Test (CRT; [5]), the extended seven-item version [32]. CRT measures the tendency to override a prepotent response alternative that is incorrect and to engage in further reflection that leads to the correct response. An item example is “A bat and a ball cost $1.10 in total. The bat costs a dollar more than the ball. How much does the ball cost?” For this question, the intuitive answer is 10 cents, but the correct answer is 5 cents. Scores range from 0 to 7, and higher scores reflect a higher ability to engage in reflective processing. CRT was successfully used in previous studies on Romanian samples (e.g., ref. [33]). On our data, the measure had acceptable reliability (α = 0.77).

Thinking styles were measured with the Rational-Experiential Inventory (REI, [14]). REI has 40 items that participants have to rate on a scale from 1 (Definitely False) to 5 (Definitely True). There are 20 items that refer to rationality (e.g., “I have no problem in thinking things through clearly.”), and 20 items that refer to experientiality (e.g., “When it comes to trusting people, I can usually rely on my gut feelings.”). Higher scores reflect a higher tendency to process information in a rational/experiential way. REI was also used in other studies on Romanian samples (e.g., ref. [34]). In our sample, both scales proved to be reliable (α = 0.90 for Rational thinking style; and α = 0.89 for Experiential thinking style). 

Attitudes toward vaccination were measured with a recently adapted instrument by ref. [35] from prior work [36]. We adapted it to the Romanian language using the standard back-translation technique. There were 10 items, such as “Some vaccines are unnecessary since they target relatively harmless diseases”, or “Vaccines are a major advancement for humanity”. Participants had to rate each item on a scale from 1 (Completely Disagree) to 5 (Completely Agree). Higher scores reflect more positive attitudes towards vaccines in general. The measure was reliable on our data (α = 0.89).

Vaccination against COVID-19 was measured via a single item: “Did you get the vaccine against COVID-19?”, with a yes or no response. Prior to this question, the participants had to state if they had the possibility of receiving the vaccine. Only those who confirmed they had the possibility were included in the main analyses (hypotheses testing). In Romania, where the present study was conducted, the vaccination against COVID-19 started on 27 December 2020. The process was divided into three stages. Within the first stage, the vaccine was administered for the medical personnel; within the second phase, it was administered for the population at risk; within the third phase, it was offered to the rest of the population. The third phase began on the 9 March 2021, at first for the cities that had a high incidence of infection and then gradually in all cities. The data for this study were collected between the 28 April and 17 May 2021. Thus, vaccination was already accessible for most of the population.

### 2.4. Statistical Approach

The first two hypotheses (H1 and H2) were tested through binomial logistic regressions implemented in IBM SPSS, and we used model 4 from the PROCESS custom dialog [37] to test the hypothesized mediation model (H3a, H3b, H3c). The indirect effects were estimated with 95% confidence intervals generated through 5000 bootstrap samples and were expressed in a log-odds metric. We introduced age in all models as a covariate, and when testing the REI factors as predictors, we also controlled for the effect of the opposite factor.

## 3. Results

### 3.1. Preliminary Analyses

Out of the recruited group of 274 participants, 217 declared that they had the possibility of becoming vaccinated against COVID-19. This sample was further used for the main analyses. Their mean age (M = 24.58, SD = 8.31) and sex distribution (53% female) resembled the entire sample closely. There were also no significant differences between those who had the chance to be vaccinated and those who did not, in terms of age (t(270) = 1.19, *p* > 0.05) and sex (χ^2^(1) = 0.04, *p* > 0.05). The majority of those who declared they had the possibility to become vaccinated (n = 217) did not receive the vaccine (n = 130; 59.9%).

Table 1 displays the correlation matrix, as well the descriptive statistics for all the studied variables. Since all the numeric variables were symmetrically distributed (skewness ranged between 0.081 and 0.535), we used the Pearson correlation to estimate the relationships between the predictors. We observed significant relations between COVID-19 vaccination behavior and cognitive reflection ability (r = 0.19, *p* = 0.005), vaccination attitudes (r = 0.43, *p* < 0.001), and participants’ age (r = 0.15, *p* < 0.05). The expected correlation between rational thinking style and vaccination behavior was not significant (r = 0.05, *p* > 0.05). Moreover, experientiality correlated with vaccination attitudes (r = −0.15, *p* < 0.05). Hence, except for the effect between rational thinking and behavior, the other relations aligned with our expectations. Moreover, the significant correlation between the dependent variable and age offers support for our decision to further keep it constant in the regression models.

#### 3.1.1. Hypotheses Testing

The results of the logistic regression analysis investigating the main effects (Table 2) showed that the probability of receiving the vaccine significantly increased with age (OR = 1.06, 95% CI: [1.02, 1.10]). This effect being held constant, higher scores on cognitive reflection ability predicted significantly higher odds of receiving the vaccine (OR = 1.18, 95% CI: [1.05, 1.36]), while the rational thinking style did not. In other words, self-reported rational thinking did not impact actual vaccination; rather, the more concretely measured ability to engage in reflective processing increased the probability of receiving the COVID-19 vaccine. Thereby, the data support H1, but not H2.

Furthermore, the experiential thinking style predicted significantly more negative vaccination attitudes (b = −0.10, *p* < 0.05), which is in alignment with hypothesis H3a. Additionally, as H3b predicted and already mentioned, the odds of receiving the vaccine increased as participants expressed more positive attitudes toward vaccination (OR = 1.20, 95% CI: [1.13, 1.27]). Finally, as can be seen in Table 3, experientiality indirectly predicted lower odds of becoming vaccinated through its negative impact on attitudes (Log Odds = −0.02, SE = 0.01, 95% CI: [−0.04, −0.001]). Hence, the last hypothesis (H3c) also gained support, and since experientiality was not directly related to vaccination behavior, the indirect relation can be classified as an indirect-only mediation effect [38].

#### 3.1.2. Supplementary Analyses

We also inspected the classification accuracy of the regression model encompassing the direct effects of the studied predictors in relation to COVID-19 vaccination behavior (Table 2), based on two indicators: the Hosmer–Lemeshow (H-L) test of goodness-of-fit and the classification accuracy of the model. A well-fitting model is indicated by a non-significant H-L chi-square statistic, a result which we found in our case (χ^2^(8) = 6.06, *p* = 0.641). Furthermore, the overall correct classification percentage was 73.5%; the model is biased towards correctly identifying those who refrained from becoming vaccinated (82.9%). 

Supplementarily, we explored two alternative mediation models: the indirect effects of cognitive reflection ability and rational thinking style in relation to vaccination behavior via attitudes. These analyses showed that neither one of the predictors had the same indirect effect on vaccination odds (see Table 3).

## 4. Discussion

The aim of the present study was to investigate the potential link between individual differences in information processing and vaccination behavior. The results indicated that cognitive reflection ability predicts vaccination behavior (H1), while a rational thinking style is unrelated to vaccination behavior (H2). However, an experiential thinking style indirectly predicts lower odds of becoming vaccinated through its negative impact on attitudes (H3). 

A better ability to override a prepotent response alternative increases the chances of deciding to become vaccinated; this means that the decision to receive the COVID-19 vaccine is more likely to happen if people are able to override their fear of potential side effects, fear which may be exacerbated by the anti-vaccination messages that are spread in the media [39]. Pro-vaccine messages include information about the COVID-19 vaccine efficacy and benefits, usually expressed in probabilistic terms [40], which require a cognitive reflection processing (system 2) to conclude that vaccination is the safer option. It would be interesting for future studies to investigate the link between cognitive reflection ability and health-related prevention behaviors but using a verbal cognitive reflection test [41]. This nonmathematical version would allow excluding potential confounding variables such as mathematical anxiety or numeracy. The lack of a relation between the tendency to approach things rationally as a thinking style and COVID-19 vaccination can be due to several factors. First, thinking styles are manifested within the domain of what is termed in the literature as type 2 processing [2]. According to ref. [2], individual differences in rational action can arise either because of individual differences in the ability to engage in reflective processing or because of individual differences in thinking dispositions, for instance, a rational thinking style. In the case of vaccination against COVID-19, it seems that the former is relevant but not the latter. Second, it is possible that the decision to vaccinate is not a clear-cut analytic task that would prompt people to employ a rational thinking approach. Ref. [42] proposed that decision tasks could be classified on a cognitive continuum ranging from intuitive to analytic and that a task’s location on the continuum determines the type of information processing that is triggered by it. Typical analytic tasks involve, for instance, abstract problems that can be solved through logical inference [43], or a solution that may be reached by following specific rules [44], which is different from the real-life decision to become vaccinated. According to ref. [13], a thinking style best predicts decision behavior when the task matches the theoretical strengths of the thinking style. Vaccination might constitute an emotionally significant context and, as the authors of ref. [16] explain, people might employ their analytic reasoning in investigating vaccination and reflect on it, but finally make the choice of avoidance based on emotional information. The emotional reaction of vaccine avoidance can be overridden only if one has a high ability of cognitive decoupling, that is, the ability to distinguish supposition from belief. Third, it is possible that specific beliefs of the population included in the present study might be at play. For instance, perceived health threat severity might be a barrier for vaccine uptake [45]. This might be especially relevant when discussing COVID-19 vaccination in young adults because the COVID-19 virus has been especially dangerous for older adults [45].

The fact that experientiality predicts lower odds of becoming vaccinated through its negative impact on attitudes is in line with recent theoretical and empirical studies (e.g., refs. [16,20]). A potential explanation for these findings is that people with an experiential style develop their attitudes based on the unpleasant emotions of worry or fear evoked by the perceived side effects. Moreover, as the authors of ref. [20] point out, in many countries (and this applies to Romania), vaccination coverage for vaccine-preventable diseases is high, which means that people cannot refer to a personal, negative experience of such a disease. However, the experience of vaccination side effects is something more prevalent. Thus, because experientiality is an affective processing mode that relies on prior experience, this asymmetry in experiences that people can refer to (sparse vaccine-preventable diseases vs. more prevalent side effects of vaccines) might explain why experientiality predicts lower odds of becoming vaccinated, through its negative impact on attitudes.

### 4.1. Practical Implications

Taken together, the present findings can help us discern the individual differences in information processing that underlie vaccination behavior against COVID-19. In turn, this understanding can inform the design of campaigns encouraging vaccine uptake. If some people base their decision to (not) vaccinate on faulty criteria primarily related to emotions [46], then we can consider this as a specific information need and address it as such, along with the need to consider literacy and unfamiliarity with scientific terminology. Following the recommendations provided by ref. [39], health authorities designing campaigns could consider the delivery preferences of target populations to ensure messages are accessible and acceptable. For instance, people that can engage in type 2 processing may respond well to messages that present information about vaccine efficacy and benefits, expressed in probabilistic terms. On the other hand, people that tend to process information in an experiential style might respond well to narrative messages about experiences of infectious disease or to messages such as “4 out of 5 people are vaccinated”—which evokes a general heuristic rule similar to social norms: if many others are doing it, it must be good (see refs. [36,47] for further examples).

### 4.2. Limitations and Future Research 

The main limitation of the present study is the cross-sectional nature of the investigation. This prevents us from demonstrating the potential causal effect of information processing type on attitudes and behavior. Future research could address this by manipulating cognitive load [48], time pressure [49], or by priming participants to use a certain type of thinking style [50] when they receive pro-vaccine messages in order to verify the effect of this manipulation on their intentions and attitudes. Another limitation refers to the sample, which was mainly composed of psychology students. This implies that the present findings can be extrapolated to populations with a similar background (higher education, young adults), but only with caution to elderly populations (the mean age of the current sample was 24), or populations with a different educational background (where vaccination motives and cognitive processing dynamic can differ). Further research is necessary to confirm if the present findings are representative of the general population. Additionally, the data for the present study were collected before any rules and laws were put into place (i.e., vaccination proof for traveling abroad, participating in any activities that took place in closed spaces, like workshops, concerts, dinner at a restaurant, or shopping at a mall). The predictors identified in this study might be overridden if the decision to become vaccinated or not has immediate pragmatic consequences (e.g., traveling abroad, participating in activities that take place in closed spaces, etc.). The present results stand for situations where no external, pragmatic constraints are involved in the decision-making process. Since evidence is beginning to accumulate on the topic of individual differences and vaccination in the context of COVID-19, future work should consider at this literature by means of a systematic review. This would allow for a structured summarization of the existing evidence, yielding future research questions that need to be addressed and more robust practical recommendations.

## 5. Conclusions

The adequacy of the messages that encourage vaccination is one of the essential stakes at play when confronting a pandemic as the one caused by COVID-19, when effective vaccines become available.

Individual differences in information processing are linked to variations in behaviors and attitudes, and this could be accounted for by tailoring messages to the specificities of the target populations.

We can devise better means of achieving our common goals by becoming acquainted with our cognitive functioning and how a specific processing leads to certain attitudes and behaviors. 

## Figures and Tables

**Figure 1 healthcare-10-01302-f001:**
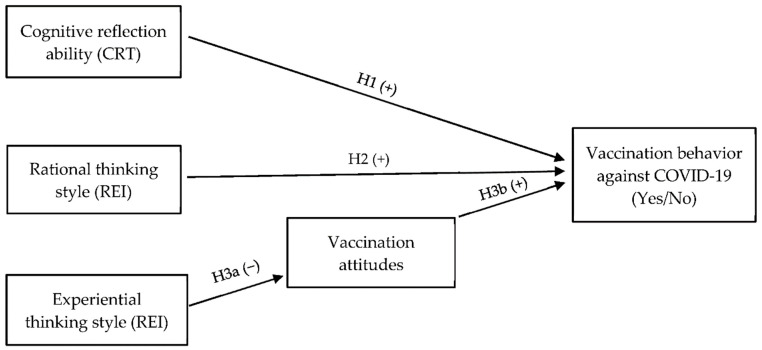
The hypothesized direct and indirect prediction models.

**Table 1 healthcare-10-01302-t001:** Descriptive statistics and Pearson correlations among the studied variables.

	*M*	*SD*	1	2	3	4	5	6
Sex	2	-	-					
Age	24.58	8.31	−0.26 **	-				
Cognitive reflection	2.62	2.20	−0.21 **	0.003	-			
Rational thinking style	3.72	0.53	0.002	−0.01	0.18 **	-		
Experiential thinking style	3.29	0.58	0.11	−0.10	−0.14 *	0.02	-	
Vaccination attitudes	3.59	0.69	−0.02	−0.07	0.12	0.07	−0.15 *	-
Vaccination	0	-	−0.10	0.15 *	0.19 **	0.05	−0.07	0.43 **

Note. *N* = 217; for the dummy variables, the mode values are displayed instead of means (Sex: 1 = male/2 = female; Vaccination: 0 = not vaccinated/1 = vaccinated). * *p* < 0.05, ** *p* < 0.01 (two-tailed).

**Table 2 healthcare-10-01302-t002:** Binomial logistic regression predicting the likelihood of receiving the COVID-19 vaccine based on age, cognitive reflection ability, rationality, experientiality, and attitudes towards vaccination.

Variable	*B*	Standard Error	Wald	*p*	Odds Ratio	95% Confidence Interval
Age	0.06	0.02	8.27	<0.01	1.06	[1.02, 1.10]
Cognitive reflection	0.17	0.07	5.08	<0.05	1.18	[1.02, 1.37]
Rational thinking style	0.01	0.02	0.10	>0.05	0.99	[0.97, 1.03]
Experiential thinking style	−0.01	0.02	0.22	>0.05	1.01	[0.98, 1.04]
Vaccination attitudes	0.18	0.03	32.86	<0.001	1.20	[1.13, 1.27]

Note: *N* = 217.

**Table 3 healthcare-10-01302-t003:** Indirect effects from experientiality, rationality, and cognitive reflection ability towards vaccination behavior, via vaccination attitudes.

Predictor	Indirect Effect	Standard Error	95% Confidence Interval	
			Lower Limit	Upper Limit
*Main model*				
Experiential thinking style	−0.02	0.01	**−0.04**	**−0.001**
*Alternative models*				
Cognitive reflection	0.06	0.04	−0.003	0.14
Rational thinking style	0.01	0.01	−0.01	0.03

Note. *N* = 217; The effect of age was controlled in each model. In the case of experientiality and rationality the opposite factor was also introduced as a covariate besides age. The significant indirect effect is highlighted in boldface. The number of bootstrap samples for the 95% confidence interval was 5000.

## Data Availability

The data presented in this study are available on request from the corresponding author.

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
