# Peer review of "I Think I Should Get Vaccinated, I Feel I Should Not. Individual Differences in Information Processing and Vaccination Behavior (COVID-19)"

_healthcare, 2022, doi:10.3390/healthcare10071302_

Round 1

Reviewer 1 Report

This is an interesting study investigating the role of cognitive reflection ability and thinking styles in predicting self-reported vaccination behavior against COVID-19. The paper is promising. But I have several concerns that require the authors' attention:

1. Page 2 last paragraph. Page 3 first paragraph.

In the last paragraph on page 2, the two thinking styles introduced are reflective thinking and experiential thinking. However on page 3 first paragraph, rational thinking was mentioned. Similarly, H2 mentions rational thinking style. If I’m not mistaken, the authors used the terms reflective thinking and rational thinking interchangeably here. The authors should try to stick to the same terminology or explicate that these two terms are interchangeable. 

2. The authors should also define reflective/rational thinking more clearly, especially considering it is the key construct of the study.

3. Page 3 third paragraph

“Intuitive or experiential thinking implies that people rely on their "gut" feeling, which means that they can be seized by their negative emotions.”

This definition is not cited, and appears slightly different to previously introduced notion of experiential thinking—holistically approaching the issue:

“…experiential thinking will likely approach problems holistically and see multiple associations between concepts.” (Page 3, 2nd line from the top)

4. Page 3 third paragraph

I think it would be good to illustrate, at least briefly, how experiential thinking is linked to negative vaccine attitudes/vaccine conspiracy beliefs specifically.  Said differently, the authors established how experiential thinking and attitudes/emotional appraisal might be link. However, what is no clear is why experiential thinking does not lead to more positive appraisals. In relation to the definition given on experiential thinking, I may have a strong “gut” feeling that positively favours being vaccinated.

5. Hypothesis H3c is slightly different from the argument the authors are proposing.

To wit, the argument the authors posited are of effects in one direction: experiential thinking leads to negative vaccination behaviour, by means of negative attitudes towards vaccination. H3c does not capture the direction of said relationship.

6. With that said, and relating to the previous point, the authors need further justification as to why the posited relationship may not exist in a positive direction. That is, experiential thinking leading to positive vaccination behaviour, by means of positive attitudes towards vaccination.

7. It would be good if the authors could provide an example of a CRT item (as did they authors do for the Rational-Experiential Inventory)

8. The examples of items should be done within quotation marks.

9. The authors mentioned that the indirect relation can be classified as a total mediation effect. However, the author could do without this part of their statement, in light of criticism against postulations of a total effect (See Zhao, Lynch, & Chen, 2010).

Zhao, X., Lynch Jr, J. G., & Chen, Q. (2010). Reconsidering Baron and Kenny: Myths and
truths about mediation analysis. Journal of Consumer Research, 37(2), 197-206.

10. Page 8, Paragraph 3.

According to [2], individual differences in rational action can arise either because of individual differences in the ability to engage in reflective processing, or because of individual differences in thinking dispositions, for instance a rational thinking style. In the case of vaccination against COVID-19, it seems that the former is relevant, but not the latter.

The argument that rational thinking is not relevant when considering vaccine uptake is weak, especially when considering that studies have shown individuals using rational justification both for and against vaccination uptake. As such, what is more likely is perhaps that between individuals, different rational reasonings were used to justify support for and against vaccine uptake, resulting in the non-significant relationship being observed.

I refer the authors to a resource that discusses cognitive barriers towards COVID-19 uptake, which, opposing to the authors postulations, makes suggestion on how rational thought might lead to vaccine hesitation (Chia et al., 2021). This should be discussed further in the manuscript

Chia et al. (2021). Cognitive barriers to COVID-19 vaccine uptake among older adults. Frontiers in Medicine, 2025. https://doi.org/10.3389/fmed.2021.756275

11. In table 1, some of the values are misaligned (forth column from the left), and the variables on the left most column should be numerated. 

12. Page 8, Paragraph 3.

… thinking dispositions, for instance a rational thinking style.

 A comma after “for instance” is needed.

Reviewer 2 Report

What proportion as a rough % of people able to be vaccinated chose not to get vaccinated? 

There seems to be a lot of work already published in this field perhaps the paper needs to be re-submitted as a meta-analysis to boost numbers and to accurately represent a cross-section of society. 

While the methods and analysis seem fine the sample size is on the small side and homogeneous - Psychology undergraduates (?). This is mentioned as a shortcoming. I hypothesize that this population section was strongly biased towards advanced cognitive reflection. It would be interesting to compare science and art students. 

A wider study on this age group comparing people who have gone to university with those who went straight into a job from school would widen the scope of the paper sufficiently - if the appropriate additional data can be accessed online. People change a lot in their early 20s - I don't know the underlying neurobiological detail this is experiential learning on my part but some kind of cognitive maturation process occurs in the early 20s  - this is an interesting age to study per se.    I am not a statistician but with data from only 217  participants going forward to the statistical analysis how can a 'bootstrap'  analysis incorporate 5000 readings? 

" The number of bootstrap samples for the 95% confidence interval was 5000."

The lack of external constraints on the decision-making processes of participants mentioned in the "strengths and weaknesses" section of the study was a strong point of the experimental design so the timing of the study on the trajectory of the pandemic was appropriate. 

English grammar needs attention throughout 

Reviewer 3 Report

In this paper, as I see, the authors have considered only undergraduate students.  One can say that the methods or analysis can be good but the sample size is very homogeneous and I am not sure that this is enough to say something about the general case. (There are only 217 participants.)  

Moreover,  I see that the terms "reflective thinking" and "rational thinking" are used as they are the same. There is confusion about it. And look at the table 1, It seems like it is formed recklessly.  

If the authors can fix all of the issues, we can re-consider the article.  

Sincerely.

Round 2

Reviewer 1 Report

The authors have adequately addressed all my comments. I appreciate all their efforts

Author Response

Thank you.

Reviewer 2 Report

I have read the responses to my comments made on the initial submission and feel that the authors have addressed my concerns adequately 

Author Response

Thank you.

Reviewer 3 Report

Dear Editor;

The authors have successfully incorporated my points and the paper in its present form is appropriate for publication in this journal. 

Sincerely Yours.

Author Response

Thank you.